# Taxonomy of Benchmarks in Graph Representation Learning

**Renming Liu**[*]
Michigan State University
liurenmi@msu.edu

**Semih Cantürk**[*]
Mila, Université de Montréal
semih.canturk@mila.quebec

**Frederik Wenkel**
Mila, Université de Montréal

**Sarah McGuire**
Michigan State University

**Xinyi Wang**
Michigan State University

**Anna Little**
University of Utah

**Leslie O'Bray**
ETH Zürich

**Michael Perlmutter**[†]
University of California, Los Angeles

**Bastian Rieck**[†]
Helmholtz Munich & TU Munich

**Matthew Hirn**[†]
Michigan State University

**Guy Wolf** [†]
Mila, Université de Montréal
wolfguy@mila.quebec

**Ladislav Rampášek**[*,†]
Mila, Université de Montréal
ladislav.rampasek@mila.quebec

## Abstract

Graph Neural Networks (GNNs) extend the success of neural networks to graph-structured data by accounting for their intrinsic geometry. While extensive research has been done on developing GNN models with superior performance according to a collection of graph representation learning benchmarks, it is currently not well understood what aspects of a given model are probed by them. For example, to what extent do they test the ability of a model to leverage graph structure vs. node features? Here, we develop a principled approach to taxonomize benchmarking datasets according to a *sensitivity profile* that is based on how much GNN performance changes due to a collection of graph perturbations. Our data-driven analysis provides a deeper understanding of which benchmarking data characteristics are leveraged by GNNs. Consequently, our taxonomy can aid in selection and development of adequate graph benchmarks, and better informed evaluation of future GNN methods. Finally, our approach and implementation in GTaxoGym package[1] are extendable to multiple graph prediction task types and future datasets.

## 1 Introduction

Machine learning for graph representation learning (GRL) has seen rapid development in recent years [29]. Originally inspired by the success of convolutional neural networks in regular Euclidean domains, thanks to their ability to leverage data-intrinsic geometries, classical graph neural network (GNN) models [16, 38, 60] extend those principles to irregular graph domain. Further advances in the field have led to a wide selection of complex and powerful GNN architectures. Some models are provably more expressive than others [46, 67], can leverage multi-resolution views of graphs [44], or can account for implicit symmetries in graph data [9]. Comprehensive surveys of graph neural networks can be found in Bronstein et al. [8], Wu et al. [65], Zhou et al. [71].

Most graph-structured data encode information in *graph structures* and *node features*. The structure of each graph represents relationships (i.e., edges) between different nodes, while the node features represent quantities of interest at each node. For example, in citation networks, nodes represent

---

[*]Equal contribution.

[†]Equal senior author contributions.

[1]https://github.com/G-Taxonomy-Workgroup/GTaxoGym

Liu & Cantürk et al., Taxonomy of Benchmarks in Graph Representation Learning. *Proceedings of the First Learning on Graphs Conference (LoG 2022)*, PMLR 198, Virtual Event, December 9–12, 2022.

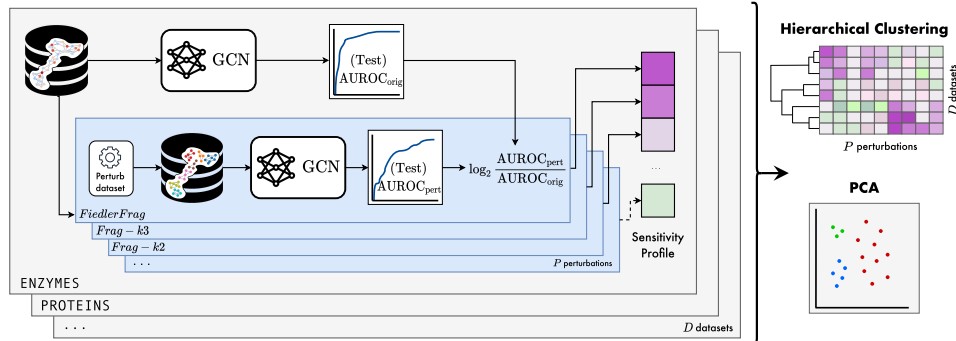

**Figure 1:** Overview of our pipeline to taxonomize graph learning datasets.

papers, and edges represent citations between the papers. On such networks, node features often capture the presence or absence of certain keywords in each paper, encoded in binary feature vectors. In graphs modeling social networks, each node represents a user, and the corresponding node features often include user statistics like gender, age, or binary encodings of personal interests.

Intuitively, the power of GNNs lies in relating local node-feature information to global graph structure information, typically achieved by applying a cascade of feature *aggregation* and *transformation* steps. In aggregation steps, information is exchanged between neighboring nodes, while transformation steps apply a (multi-layer) perceptron to feature vectors of each node individually. Such architectures are commonly referred to as *Message Passing Neural Networks (MPNN)* [25].

Historically, GNN methods have been evaluated on a small collection of datasets [47], many of which originated from the development of graph kernels. The limited quantity, size and variety of these datasets have rendered them insufficient to serve as distinguishing benchmarks [18, 49]. Therefore, recent work has focused on compiling a set of large(r) benchmarking datasets across diverse graph domains [18, 33]. Despite these efforts and the introduction of new datasets, it is still not well understood what aspects of a dataset most influence the performance of GNNs. Which is more important, the geometric structure of the graph or the node features? Are long-range interactions crucial, or are short-range interactions sufficient for most tasks? This lack of understanding of the dataset properties and of their similarities makes it difficult to select a benchmarking suit that would enable comprehensive evaluation of GNN models. Even when an array of seemingly different datasets is used, they may probe similar aspects of graph representation learning.

Leveraging symmetries and other geometric priors in graph data is crucial for generalizable learning [9]. While invariance or equivariance to some transformations is inherent, invariance to others may only be empirically or partially apparent. Motivated by this observation, we propose to use the lens of empirical transformation sensitivity to gauge *how* task-related information is encoded in graph datasets and subsequently taxonomize their use as benchmarks in graph representation learning. Our approach is illustrated in Figure 1. Namely, we list our contributions in this study as:

1. We develop a graph dataset taxonomization framework that is extendable to both new datasets and evaluation of additional graph/task properties.

2. Using this framework, we provide the first taxonomization of GNN (and GRL) benchmarking datasets, collected from TUDatasets [47], OGB [33] and other sources.

3. Through the resulting taxonomy, we provide insights about existing datasets and guide better dataset selection in future benchmarking of GNN models.

## 2   Methods

As a proxy for invariance or sensitivity to graph perturbations, we study the changes in GNN performance on perturbed versions of each dataset. These perturbations are designed to eliminate or emphasize particular types of information embedded in the graphs. We define an empirical *sensitivity profile* of a dataset as a vector where each element is the performance of a GNN after a given perturbation, reported as a percentage of the network's performance on the original dataset. In particular, we use a set of 13 perturbations, visualized in Figure 2. Of these perturbations, 6 are

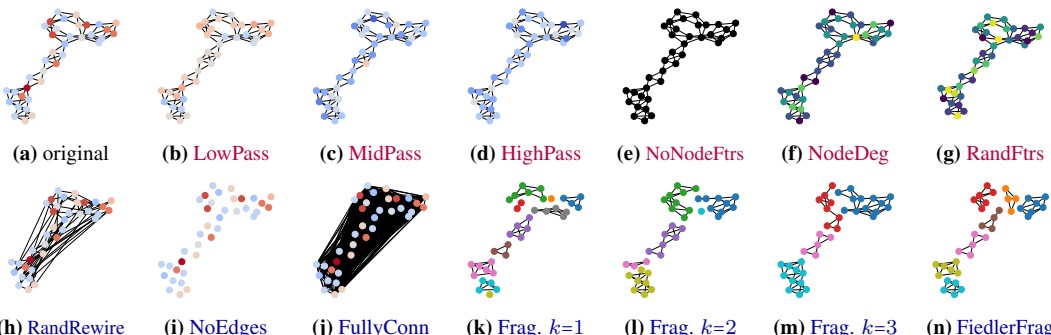

**Figure 2:** Node feature and graph structure perturbations of the first graph in ENZYMES. The color coding of nodes illustrates their feature values, except (k-n) where the fragment assignment is shown.

designed to perturb node features, while keeping the graph structure intact, whereas the remaining 7 keep the node attributes the same, but manipulate the graph structure.

For the purpose of these perturbations, we consider all graphs to be undirected and unweighted, and assume they all have node features, but not edge features. These assumptions hold for most datasets we use in this study. However, if necessary, we preprocess the data by symmetrizing each graph's adjacency matrix and dropping any edge attributes. Formally, let $G = (V, E, \mathbf{X})$ be an undirected, unweighted, attributed graph with node set $V$ of cardinality $|V| = n$, edge set $E \subset V \times V$, and a matrix of $d$-dimensional node features $\mathbf{X} \in \mathbb{R}^{n \times d}$. We let $\mathbf{M} \in \mathbb{R}^{n \times n}$ denote the adjacency matrix of each graph, where $\mathbf{M}(u, v) = 1$ if $(u, v) \in E$ and zero otherwise.

Several of our perturbations are based on spectral graph theory, which represents graph signals in a spectral domain analogous to classical Fourier analysis. We define the graph Laplacian $\mathbf{L} \coloneqq \mathbf{D} - \mathbf{M}$ and the symmetric normalized graph Laplacian $\mathbf{N} \coloneqq \mathbf{D}^{-\frac{1}{2}} \mathbf{L} \mathbf{D}^{-\frac{1}{2}} = \mathbf{I} - \mathbf{D}^{-\frac{1}{2}} \mathbf{M} \mathbf{D}^{-\frac{1}{2}}$, where $\mathbf{D}$ is the diagonal degree matrix. Both $\mathbf{L}$ and $\mathbf{N}$ are positive semi-definite and have an orthonormal eigendecompositions $\mathbf{L} = \mathbf{\Phi} \mathbf{\Lambda} \mathbf{\Phi}^\top$ and $\mathbf{N} = \tilde{\mathbf{\Phi}} \tilde{\mathbf{\Lambda}} \tilde{\mathbf{\Phi}}^\top$. By convention, we order the eigenvalues and corresponding eigenvectors $\{(\lambda_i, \phi_i)\}_{0 \leq i \leq n-1}$ of $\mathbf{L}$ (and similarly for $\mathbf{N}$) in ascending order $0 = \lambda_0 \leq \lambda_1 \leq \cdots \leq \lambda_{n-1}$. The eigenvectors $\{\phi_i\}_{0 \leq i \leq n-1}$ constitute a basis of the space of graph signals and can be considered as generalized Fourier modes. The eigenvalues $\{\lambda_i\}_{0 \leq i \leq n-1}$ characterize the variation of these Fourier modes over the graph and can be interpreted as (squared) frequencies.

## 2.1 Node Feature Perturbations

We first consider two perturbations that alter local node features, setting them either to a fixed constant (w.l.o.g., one) for all nodes, or to a one-hot encoding of the degree of the node. We refer to these perturbations as *NoNodeFtrs* (as constant node features carry no additional information) and *NodeDeg*, respectively. Sensitivity to these perturbations, exhibited by a large decrease in predictive performance, may indicate that a task is dominated by highly informative node features. Further, we consider a random node feature perturbation (*RandFtrs*) by sampling a one-dimensional feature for each node from $\mathcal{U}_{[-1,1]}$, which has been shown to improve the WL expressiveness of MPNNs [1, 54].

We also develop spectral node feature perturbations. As in Euclidean settings, the Fourier decomposition can be used to decompose graph signals into a set of canonical signals, called Fourier modes, which are organized according to increasing variation (or frequency). In Euclidean Fourier analysis, these modes are sinusoidal waves oscillating at different frequencies. A standard practice in audio signal processing is to remove noise from a signal by identifying and removing certain Fourier modes or *frequency bands*. We generalize this technique to graph datasets and systematically remove certain graph Fourier modes to probe the importance of the corresponding frequency bands.

In this perturbation, we use the frequencies derived from the symmetric normalized graph Laplacian $\mathbf{N}$ and split them into three roughly equal-sized frequency bands (*low*, *mid*, *high*), i.e., bins of subsequent eigenvalues. To assess the importance of each of the frequency bands, we then apply *hard* band-pass filtering to the graph signals (node feature vectors), i.e., we project the signals on the span of the selected Fourier modes. More specifically, for each band, we let $\mathbf{I}_{\text{band}}$ be a diagonal matrix with diagonal elements equal to one if the corresponding eigenvalue is in the band, and zero otherwise. Then, the hard band-pass filtered signal is computed as

$$\mathbf{X}_{\text{band}} = \tilde{\boldsymbol{\Phi}} \mathbf{I}_{\text{band}} \tilde{\boldsymbol{\Phi}}^\top \mathbf{X}. \tag{1}$$

The above band-pass filtering perturbation enables a precise selection of the frequency bands. However, it requires a full eigendecomposition of the normalized graph Laplacian, which is impractical for large graphs. We therefore provide an alternative approach based on wavelet bank filtering [13]. This leverages the fact that polynomial filters $h$ of the normalized graph Laplacian directly transform the spectrum via $h(\mathbf{N}) = \tilde{\boldsymbol{\Phi}} h(\tilde{\boldsymbol{\Lambda}}) \tilde{\boldsymbol{\Phi}}^\top$, yielding the *frequency response* $h(\lambda)$ for any eigenvalue $\lambda$ of $\mathbf{N}$. This is usually done by taking the symmetrized diffusion matrix

$$\mathbf{T} = \frac{1}{2} (\mathbf{I} + \mathbf{D}^{-\frac{1}{2}} \mathbf{M} \mathbf{D}^{-\frac{1}{2}}) = \frac{1}{2} (2\mathbf{I} - \mathbf{N}). \tag{2}$$

By construction, $\mathbf{T}$ admits the same eigenbasis as $\mathbf{N}$ but its eigenvalues are mapped from $[0, 2]$ to $[0, 1]$ via the frequency response $h(\lambda) = 1 - \lambda/2$. As a result, large eigenvalues are mapped to small values (and vice versa). Next, we construct *diffusion wavelets* [16] that consist of differences of dyadic powers $2^k, k \in \mathbb{N}_0$ of $\mathbf{T}$, i.e., $\Psi_k = \mathbf{T}^{2^{k-1}} - \mathbf{T}^{2^k}$, which act as bandpass filters on the signal. Intuitively, this operator "compares" two neighborhoods of different sizes (radius $2^{k-1}$ and $2^k$) at each node. Diffusion wavelets are usually maintained in a wavelet bank $\mathcal{W}_K = \{ \boldsymbol{\Psi}_k, \boldsymbol{\Phi}_{\mathbf{K}} \}_{k=0}^K$, which contains additional highpass $\boldsymbol{\Psi}_0 = \mathbf{I} - \mathbf{T}$ and lowpass $\boldsymbol{\Psi}_{\mathbf{K}} = \mathbf{T}^K$ filters. In our experiments, we choose $K = 1$, resulting in the following low, mid, and highpass filtered node features:

$$\mathbf{X}_{\text{high}} = (\mathbf{I} - \mathbf{T})\mathbf{X}, \quad \mathbf{X}_{\text{mid}} = (\mathbf{T} - \mathbf{T}^2)\mathbf{X}, \quad \mathbf{X}_{\text{low}} = \mathbf{T}^2 \mathbf{X}. \tag{3}$$

These filters correspond to frequency responses $h_{\text{high}}(\lambda) = \lambda/2$, $h_{\text{mid}}(\lambda) = (1 - \lambda/2) - (1 - \lambda/2)^2$ and $h_{\text{low}}(\lambda) = (1 - \lambda/2)^2$. Therefore, the low-pass filtering preserves low-frequency information while suppressing high-frequency information, whereas high-pass filtering does the opposite. The mid-pass filtering suppresses all frequencies. However, it preserves much more middle-frequency information than it does high- or low-frequency information.

Therefore, this filtering may be interpreted as an approximation of the hard band-pass filtering discussed above. From the spatial message passing perspective, low-pass filtering is equivalent to local averaging of the node features, which has a profound implication on homophilic and heterophilic characteristics of the datasets (Sec. 3.2). Finally, since the computations needed in (3) can be carried out via sparse matrix multiplications, they scale much better to large graphs. Therefore, we utilize the wavelet bank filtering for the datasets with larger graphs considered in Sec. 3.2, while for the smaller graphs, considered in Sec. 3.1, we employ the direct band-pass filtering approach.

## 2.2 Graph Structure Perturbations

The following perturbations act on the graph structure by altering the adjacency matrix. By removing all edges (*NoEdges)* or making the graph fully-connected (*FullyConn*), we can eliminate the structural information completely and essentially turn the graph into a set. The difference between the two perturbations lies in whether all nodes are processed independently or together. However, *FullyConn* is only applied to inductive datasets in Sec. 3.1 due to computational limitations. Furthermore, we consider a degree-preserving random edge rewiring perturbation (*RandRewire*). In each step, we randomly sample a pair of edges and randomly exchange their end nodes. We then repeat this process without replacement until 50% of the edges have been randomly rewired.

To inspect the importance of local vs. global graph structure, we designed the *Frag-k* perturbations, which randomly partition the graph into connected components consisting of nodes whose distance to a seed node is less than $k$. Specifically, we randomly draw one seed node at a time and extract its $k$-hop neighborhood by eliminating all edges between this new fragment and the rest of the graph; we repeat this process on the remaining graph until the whole graph is processed. A smaller $k$ implies smaller components, and hence discards the global structure and long-range interactions.

Graph fragmentations can also be constructed using spectral graph theory. In our taxonomization, we adopt one such method, which we refer to as Fiedler fragmentation (*FiedlerFrag*) (see [36] and the references therein). In the case when the graph $G$ is connected, $\phi_0$, the eigenvector of the graph Laplacian $\mathbf{L}$ corresponding to $\lambda_0 = 0$, is constant. The eigenvector $\phi_1$ corresponding to the next smallest eigenvalue, $\lambda_1$, is known as the *Fiedler vector* [22]. Since $\phi_0$ is constant, it follows that $\phi_1$ has zero average. This motivates partitioning the graph into two sets of vertices, one where $\phi_1$ is positive and the other where $\phi_1$ is negative. We refer to this process as binary Fiedler fragmentation. This heuristic is used to construct the ratio cut for a connected graph [28]. The ratio cut partitions

a connected graph into two disjoint connected components $V = U \uplus W$, such that the objective $|E(U, W)|/(|U| \cdot |W|)$ is minimized, where $E(U, W) \coloneqq \{(u, w) \in E : u \in U, w \in W\}$ is the set of removed edges when fragmenting $G$ accordingly. This can be seen as a combination of the min-cut objective (numerator), while encouraging a balanced partition (denominator).

*FiedlerFrag* is based on iteratively applying binary Fiedler fragmentation. In each step, we separate the graph into its connected components and apply binary Fiedler fragmentation to the largest component. We repeat this process until either we reach 200 iterations, or the size of the largest connected component falls below 20. In contrast to the random fragmentation *Frag-k*, this perturbation preserves densely connected regions of the graph and eliminates connections between them. Thus, *FiedlerFrag* tests the importance of *inter community message flow*. Due to computational limits, we only apply *FiedlerFrag* to inductive datasets in Sec. 3.1 for which this computation is feasible.

## 2.3 Data-driven Taxonomization by Hierarchical Clustering

To study a systematic classification of the graph datasets, we use Ward's method [62] for hierarchical clustering analysis on their *sensitivity profiles*. The *sensitivity profiles* are established empirically by contrasting the performance of a GNN model on a perturbed dataset and on the original dataset. To quantify this performance change, we use $\log_2$-transformed ratio of test AUROC (area under the ROC curve). Thus a sensitivity profile is a 1-D vector with as many elements as we have in perturbation experiments. See Figure 1 and Appendix A for further details.

In order to generate *sensitivity profiles*, we must select suitable GNN models based on several practical considerations: (i) The model has to be expressive enough to efficiently leverage aspects of the node features and graph structure that we perturb. Otherwise, our analysis will not be able to uncover reliance on these properties. (ii) The model needs to be general enough to be applicable to a wide variety of datasets, avoiding dataset-specific adjustments that may lead to profiling that is not comparable between datasets. Therefore, we did not aim for specialized models that maximize performance, but rather models that (i) achieve at least baseline performance comparable to published works over all datasets, (ii) have manageable computational complexity to facilitate large-scale experimentation, and (iii) use well-established and theoretically well-understood architectures.

With these criteria in mind, we focused on two popular MPNN models in our analysis: GCN [38] and GIN [67]. The original GCN serves as an ideal starting point as its abilities and limitations are well understood. However, we also wanted to perform taxonomization through a provably more expressive and recent method, which motivated our selection of GIN as the second architecture. We emphasize that the main focus here is not to provide a benchmarking of GNN models *per se*, but rather to address the taxonomization of *graph datasets* (and accompanying tasks) used in such benchmarks. Nevertheless, we have also generated sensitivity profiles by additional models in order to comparatively demonstrate the robustness of our approach: 2-Layer GIN, ChebNet [16], GatedGCN [7] and GCN II [11]; see Figure 5.

# 3 Results

Each of the 49 datasets we consider is equipped with either a node classification or graph classification task. In the case of node classification, we further differentiate between the *inductive* setting, in which learning is done on a set of graphs and the generalization occurs from a training set of graphs to a test set, and the *transductive* setting, in which learning is done in one (large) graph and the generalization occurs between subsets of nodes in this graph. Graph classification tasks, by contrast, always appear in an *inductive* setting. The only major difference between graph classification and inductive node classification is that prior to final prediction, the hidden representations of all nodes are pooled into a single graph-level representation. In the following two subsections, we provide an analysis of the sensitivity profiles for datasets with inductive and transductive tasks.

## 3.1 Taxonomy of Inductive Benchmarks

**Datasets.** We examine a total of 24 datasets, 21 of which are equipped with a graph-classification task (inductive by nature) and the other three are equipped with an inductive node-classification task. Of these datasets, 18 are derived from real-world data, while the other six are synthetically generated.

For real-world data, we consider several domains. Biochemistry tasks are the most ubiquitous, including compound classification based on effects on cancer or HIV inhibition (`NCI1` & `NCI109` [61],

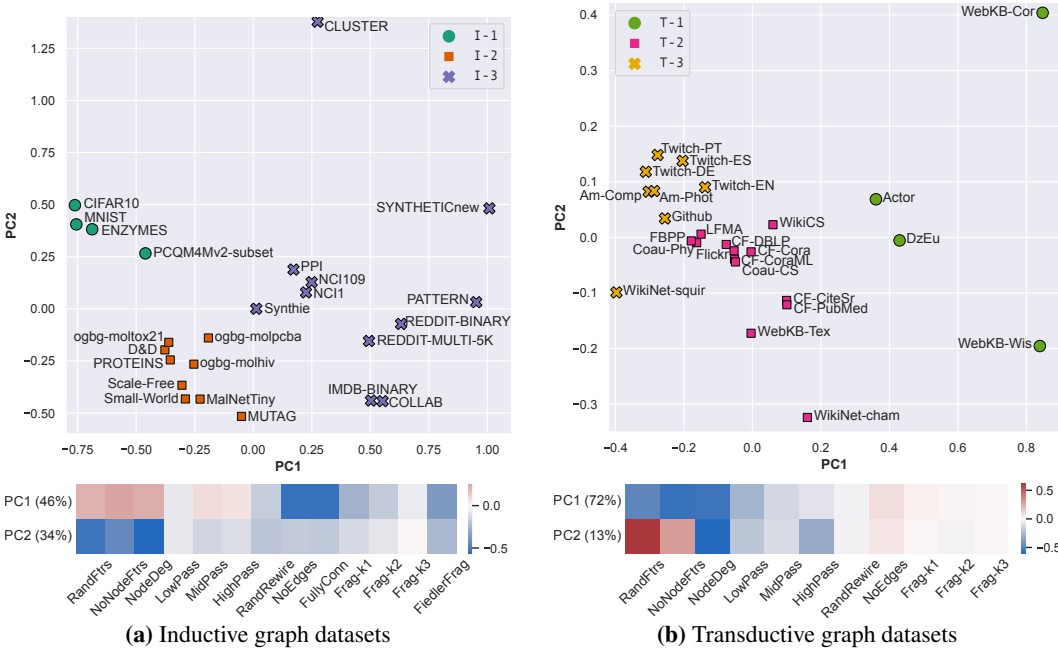

**(a)** Inductive graph datasets    **(b)** Transductive graph datasets

**Figure 3:** Visualization of (a) inductive and (b) transductive datasets based on PCA of their perturbation *sensitivity profiles* according to a GCN model. The datasets are labeled according to their taxonomization by hierarchical clustering, shown in Figure 4 and 6, which corroborates with the emerging clustering in the PCA plots. In the bottom part are shown the loadings of the first two principal components and (in parenthesis) the percentage of variance explained by each of them.

ogbg-molhiv [33]), protein-protein interaction PPI [30, 72], multilabel compound classification based on toxicity on biological targets (ogbg-moltox21 [33]), and multiclass classification of enzymes (ENZYMES [33]). We also consider superpixel-based graph classification as an extension of image classification (MNIST & CIFAR10 [18]), collaboration datasets (IMDB-BINARY & COLLAB [68]), and social graphs (REDDIT-BINARY & REDDIT-MULTI-5K [68]).

For synthetic data, we have a concrete understanding of their graph domain properties and how these properties relate to their respective prediction tasks. This allows us to derive a deeper understanding of their *sensitivity profiles*. The six synthetic datasets in our study make use of a varied set of graph generation algorithms. Small-world [69] is based on graph generation with the Watz-Strogatz (WS) model; the task is to classify graphs based on average path length. Scale-free [69] retains the same task definition, but the graph generation algorithm is an extension of the Barabási-Albert (BA) model proposed by Holme and Kim [32]. PATTERN and CLUSTER are node-level classification tasks generated with stochastic block models (SBM) [31]. Synthie [45] graphs are derived by first sampling graphs from the well-known Erdös-Rényi (ER) model, then deriving each class of graphs by a specific graph surgery and sampling of node features from a distinct distribution per each class. Similarly, SYNTHETICnew [19] graphs are generated from a random graph, where different classes are formed by specific modifications to the original graph structure and node features. Further details of dataset definitions and synthetic graph generation algorithms are provided in Appendix C.

**Insights.** Here we itemize the main insights into inductive datasets. Our full taxonomy is shown in Figures 4 and 3a, with a detailed analysis of individual clusters given in Appendix B.1.

- **Three distinct groups of datasets.** We identify a categorization into three dataset clusters I-{1,2,3} that emerge from both the hierarchical clustering and PCA. The datasets in I-{1,2} exhibit stronger node feature dependency and do not encode crucial information in the graph structure. The main differentiating factor between I-1 and I-2 is their relative sensitivity to node feature perturbations – in particular, how well *NodeDeg* can substitute the original node features. On the other hand, datasets in I-3 rely considerably more on graph structure for correct task prediction. This is also reflected by the first two principal components (Figure 3a), where PC1 approximately corresponds to structural perturbations and PC2 to node feature perturbations.

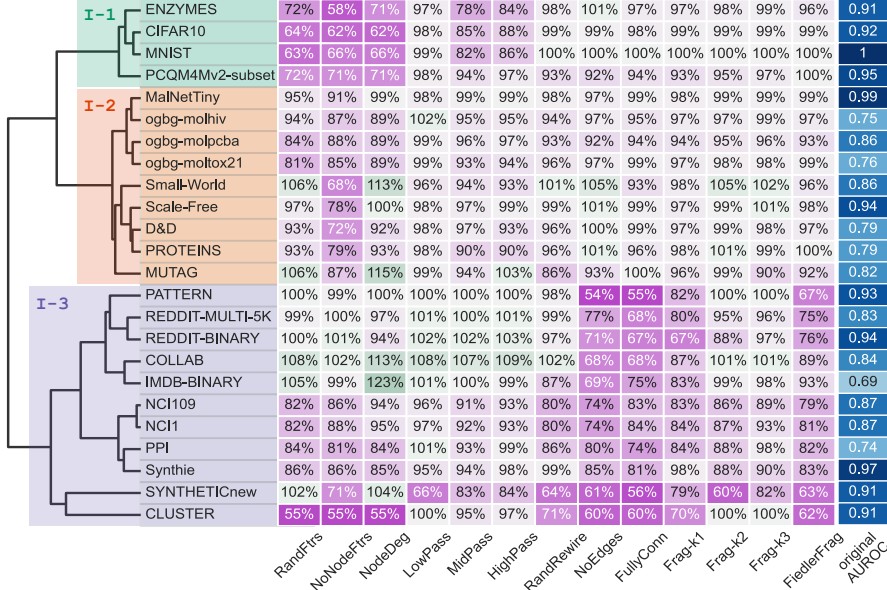

**Figure 4:** Taxonomy of inductive graph learning datasets via graph perturbations. For each dataset and perturbation combination, we show the GCN model performance relative to its performance on the unmodified dataset.

- **No clear clustering by dataset domain.** While datasets that are derived in a similar fashion cluster together (e.g., REDDIT-* datasets), in general, each of the three clusters contains datasets from a variety of application domains. Not all molecular datasets behave alike; e.g., ogbg-mol* datasets in I-2 considerably differ from NCI* datasets in I-3.

- **Synthetic datasets do not fully represent real-world scenarios.** CLUSTER, SYNTHETICnew, and PATTERN lie at the periphery of the PCA embeddings, suggesting that existing synthetic datasets do not resemble the type of complexity encountered in real-world data. Hence, one should use synthetic datasets in conjunction with real-world datasets to comprehensively evaluate GNN performance rather than solely relying on synthetic ones. We also note that the sensitivity profiles of all synthetic datasets are well-accounted for w.r.t. their respective design criteria which validate our approach; we refer the reader to Appendix B.1 for a more detailed analysis.

- **Representative set.** One can now select a representative subset of all datasets to cover the observed heterogeneity among the datasets. Our recommendation: PCQM4Mv2-subset, CIFAR10 from I-1; D&D, ogbg-molpcba from I-2; NCI1, COLLAB, REDDIT-MULTI-5K, CLUSTER from I-3.

- **Robustness w.r.t. GNN choice.** In addition to GCN, we have performed our perturbation analysis w.r.t. GIN [67], 2-Layer GIN, ChebNet [16], GatedGCN [7] and GCN II [11]. These models were selected to cover a variety of inductive model biases: GIN is provably 1-WL expressive, ChebNet uses a higher-order approximation of the Laplacian, GatedGCN employs gating akin to attention, and GCN II leverages skip connections and identity mapping to alleviate oversmoothing. We have also tested a 2-layer GIN to probe the robustness to the number of message-passing layers. The taxonomies w.r.t. other models (Figure B.1) are congruent with that of GCN. Given the differing inductive biases and representational capacity, some differences in the sensitivity

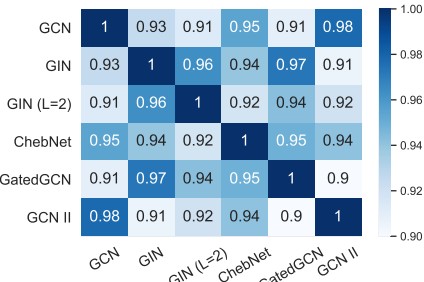

**Figure 5:** Pearson correlation between profiles derived by six GNN models.

profiles are not only expected but desired to validate their functions in benchmarking. The resulting profiles can be used for a detailed comparative analysis of these models, but the overall conclusions remain consistent. This consistency is further validated by our correlation analysis amongst these models, shown in Figure 5. The Pearson correlation coefficients of all pairs are above 90%, implying that our taxonomy is sufficiently robust w.r.t. different GNNs and the number of layers.

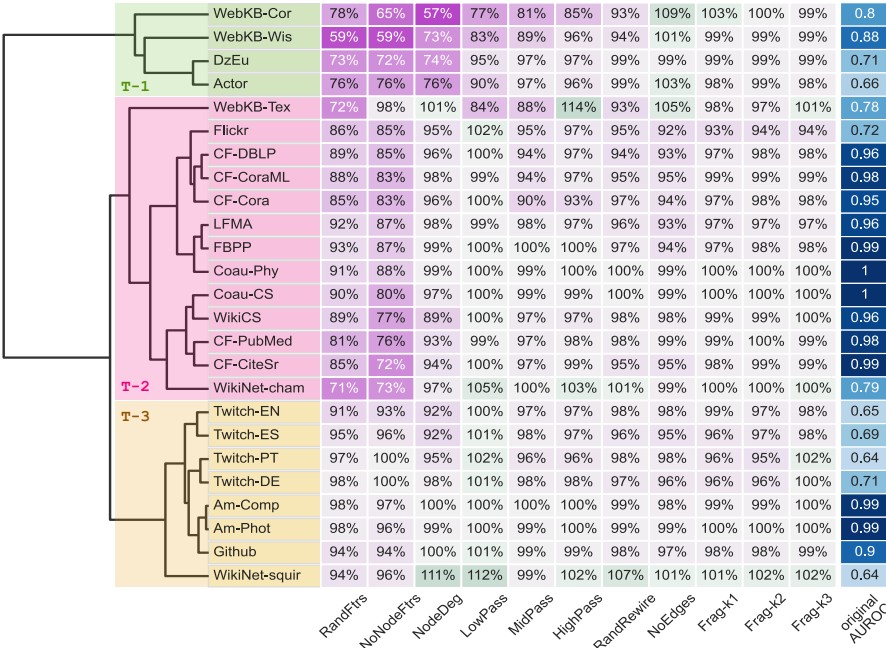

**Figure 6:** Taxonomization of transductive datasets based on sensitivity profiles w.r.t. a GCN model.

## 3.2 Taxonomy of Transductive Benchmarks

**Datasets.** We selected a wide variety of 25 transductive datasets with node classification tasks, including citation networks, social networks, and other web page derived networks (see Appendix C). In citation networks, such as CitationFull (CF) [5], nodes and edges correspond to papers that are linked via citation. In web page derived networks, like WikiNet [51], Actor [51], and WikiCS [43], they correspond to hyperlinks between pages. In social networks, like Deezer (DzEu) [53], LastFM (LFMA) [53], Twitch [52], Facebook (FBPP) [52], Github [52], and Coau [56], nodes and edges are based on a type of relationship, such as mutual-friendship and co-authorship. Flickr [70] and Amazon [56] are constructed based on other notions of similarity between entities, such as co-purchasing and image property similarities. WebKB [51] contains networks of university web pages connected via hyperlinks. It is an example of a *heterophilic* dataset [48], since immediate neighbor nodes do not necessarily share the same labels (which correspond to a user's role, such as faculty or graduate student). By contrast, Cora, CiteSeer, and PubMed are known to be *homophilic* datasets where nodes within a neighborhood are likely to share the same label. In fact, no less than 60% of nodes in these networks have neighborhoods that share the same node label as the central node [43].

**Insights.** Below we list the main insights into transductive graph datasets and their taxonomy (Figures 6 and 3b). We refer the reader to Appendix B.2 for the analysis of individual clusters.

- **Transductive datasets are uniformly insensitive to structural perturbations.** Sensitivity profiles of *all* transductive datasets show high robustness to *all* graph structure perturbations. This is in stark contrast with the inductive datasets, where the largest cluster I-3 is defined by high sensitivity to structural perturbations. The graph connectivity may not be vital to every dataset/task, e.g., in WikiCS word embeddings of Wikipedia pages may be sufficient for categorization without hyperlinks. While the observation that no dataset significantly depends on structural information is startling, it corroborates with the reported strong performance of MLP or similar models augmented with label propagation to outperform GNNs in several of these transductive datasets [24, 35].

- **Three distinct groups of datasets.** The transductive datasets are also categorized into three clusters as T-{1,2,3}. T-1 consists of heterophilic datasets, such as WebKB and Actor [42, 48]. These are well-separated from others, as seen in the right half of the PCA plot (Figure 3b), primarily via PC1 and characterized by performance drop due to removal of the original node features (*NoNodeFtrs*, *RandFtrs*) and their replacement by node degrees (*NodeDeg*). T-3 is indifferent to both node and structure removal, implying redundancies between node features and graph structure for their tasks. T-2 datasets, on the other hand, experience significant performance degradation on *NoNodeFtrs*

and *RandFtrs*, yet these drops are recovered in *NodeDeg*. This indicates that T-2 datasets have tasks for which structural summary information is sufficient, perhaps due to homophily.

- **Representative set.** Many datasets have very close sensitivity profiles, thus factoring in also the graph size and original AUROC (avoiding saturated datasets), we make the following recommendation: `WebKB-Wis`, `Actor` from T-1; `WikiNet-cham`, `WikiCS`, `Flickr` from T-2; `WikiNet-squir`, `Twitch-EN`, `GitHub` from T-3.

## 4 Discussion

Our results quantify the extent to which graph features or structures are more important for the downstream tasks; a vital question brought up in classical works on graph kernels [40, 55]. We observed that more than half of the datasets contain rich node features. On average, excluding these features reduces GNN prediction performance more than excluding the entire graph structures, especially for transductive node-level tasks. Furthermore, low-frequency information in node features appears to be essential in most datasets that rely on node features. Historically, most graph data aimed to capture closeness among entities, which has prompted the development of local aggregation approaches, such as label propagation, personalized page rank, and diffusion kernels [14, 39], all of which share a common principle of low pass filtering. High-frequency information, on the other hand, may be important in recently emerging application areas, such as combinatorial optimization, logical reasoning or biochemical property prediction, which require complex non-local representations.

Further, despite the recent interest in the development of new methods that could leverage long-range dependencies and heterophily, the availability of adequate benchmarking datasets remains lacking or less readily accessible. Meanwhile, some recent efforts, such as GraphWorld [49], aim to comprehensively profile a GNN's performance using a collection of synthetic datasets that cover an entire parametric space. Notably, our analysis demonstrates that synthetic tasks do not fully resemble the complexity of real-world applications. Hence, benchmarking made purely by synthetic datasets should be taken with caution, as the behavior might not be representative of real-world scenarios.

As a comprehensive benchmarking framework, our work provides several potential use cases beyond the taxonomy analysis presented here. One such usage is understanding the characteristics of any new datasets and how they are related to existing ones. For example, DeezerEurope (`DzEu`) is a relatively new dataset [53] that is less commonly benchmarked and studied than the other datasets we consider. The inclusion of `DzEu` in T-1 suggested its heterophilic nature, which indeed has been recently demonstrated [41]. On the other hand, since the sensitivity profiles naturally suggest the invariances that are important for different datasets from a practical standpoint, they could provide valuable guidance to the development of self-supervised learning and data augmentations for GNNs [66].

Finally, we observed that overall patterns in *sensitivity profiles* remain similar regardless of whether we used GCN, GIN, or the other 4 models to derive them. Subtle differences in sensitivity profiles w.r.t. different GNN models are not only expected but also desired when comparing models that have distinct levels of expressivity. While we expect overall patterns to be similar, more expressive models should provide enhanced resolution. One could then contrast taxonomization w.r.t. first-order GNNs (such as those we used) with more expressive higher-order GNNs, Transformer-based models with global attention, and others. We hope our work will also inspire future work to empirically validate the expressivity of new graph learning methods in this vein beyond classical benchmarking.

**Limitations and Future Work.** Our perturbation-based approach is fundamentally limited in that we cannot test the significance of a property that we cannot perturb or that the reference GNN model cannot capture. Therefore, designing more sophisticated perturbation strategies to gauge specific relations could bring further insight into the datasets and GNN models alike. New perturbations may gauge the usefulness of geometric substructures such as cycles [3] or the effects of graph bottlenecks, e.g., by rewiring graphs to modify their "curvatures" [59]. Other perturbations could include graph sparsification (edge removal) [57] and graph coarsening (edge contraction) [4, 10].

A number of OGB node-level datasets are not included in this study due to the memory cost of typical MPNNs. Conducting an analysis based on recent scalable GNN models [21] would be an interesting avenue of future research. Further, we only considered classification tasks, omitting regression tasks, as their evaluation metrics are not easily comparable. One way to circumvent this issue would be to quantize regression tasks into classification tasks by binning their continuous targets. Additionally, we disregarded edge features in two OGB molecular datasets we used. In a future work, edge

features could be leveraged by an edge-feature-aware generalization of MPNNs. The importance of edge features can then be analyzed by introducing new edge-feature perturbations. We also limited our analysis to node-level and graph-level tasks, but this framework could be further extended to link-prediction or edge-level tasks. While our perturbations could be used in this new scenario as well, new perturbations, such as graph sparsification, would need to be considered. Similarly, hallmark models for link and relation predictions, outside MPNNs, should be considered.

## 5 Conclusion

We provide a systematic data-driven approach for taxonomizing a large collection of graph datasets – the first study of its kind. The core principle of our approach is to gauge the essential characteristics of a given dataset with respect to its accompanying prediction task by inspecting the downstream effects caused by perturbing its graph data. The resulting sensitivities to the diverse set of perturbations serve as "fingerprints" that allow identifying datasets with similar characteristics. We derive several insights into the current common benchmarks used in the field of graph representation learning and make recommendations on the selection of representative benchmarking suits. Our analysis also puts forward a foundation for evaluating new benchmarking datasets that will likely emerge in the field.

## Acknowledgements

This work was partially funded by Fin-ML CREATE graduate studies scholarship for PhD [*F.W.*]; IVADO (Institut de valorisation des données) grant PRF-2019-3583139727, FRQNT (Fonds de recherche du Québec - Nature et technologies) grant 299376, Canada CIFAR AI Chair [*G.W.*]; NSF (National Science Foundation) grant DMS-1845856 [*M.H.*]; and NIH (National Institutes of Health) grant NIGMS-R01GM135929 [*M.H.,G.W.*] The content provided here is solely the responsibility of the authors and does not necessarily represent the official views of the funding agencies.

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

# A  Extended Methods

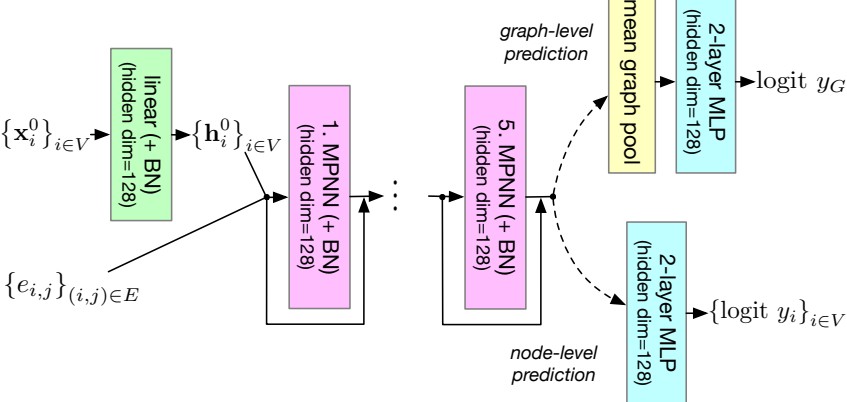

**Figure A.1:** MPNN model blueprint used for all datasets.

## A.1  Taxonomization by Hierarchical Clustering

To study a systematic classification of the graph datasets, we use Ward's method [62] for hierarchical clustering analysis on their *sensitivity profiles*. Specifically, we first construct a perturbation sensitivity matrix where each row represents a dataset and each column represents a perturbation. An entry in this matrix is computed by taking the ratio between the test score achieved with the perturbed dataset and the test score achieved with the original dataset. As our performance metric we use the area under the receiver operating characteristic (AUROC) averaged over 10 random seed runs or 10 cross-validation folds, depending on whether a dataset has predefined data splits or not. Row-wise hierarchical clustering provides us a data-driven taxonomization of the datasets.

Using AUROC as our metric, the values of the perturbation sensitivity matrix range from $0.5$ to $1$ when a perturbation causes a loss in predictive performance, and from $1$ to $2$ when it improves it. Therefore we element-wise $\log_2$-transform the matrix to balance the two ranges and map the values onto $[-1, 1]$ before hierarchical clustering. Yet, for a more intuitive presentation, we show the original ratio values as percentages throughout this paper.

## A.2  MPNN Hyperparameter Selection

We keep the model hyperparameters, illustrated in Figure A.1, identical for each dataset and perturbation combination. We use a linear node embedding layer, 5 graph convolutional layers with residual connections and batch normalization (only for inductive datasets), followed by global mean pooling (in case of graph-level prediction tasks), and finally a 2-layer MLP classifier. For training we use Adam optimizer [37] with learning rate reduction by 0.5 factor upon reaching a validation loss plateau. Early stopping is done based on validation split performance.

**Implementation.** Our pipeline is built using PyTorch [50] and PyG [20] with GraphGym [69] (provided under MIT License). Its modular & scalable design facilitated here one of the most extensive experimental evaluations of graph datasets to date.

**Computing environment and used resources.** All experiments were run in a shared computing cluster environment with varying CPU and GPU architectures. These involved a mix of NVidia V100 (32GB), RTX8000 (48GB), and A100 (40GB) GPUs. The resource budget for each experiment was 1 GPU, 4 CPUs, and up to 32GB system RAM.

# B  Extended Results

## B.1  Taxonomy of Inductive Benchmarks

`I-1`**: Node-feature reliance.**  The top-most cluster `I-1`, while mostly indifferent to structural perturbations, is highly sensitive to node feature perturbations that comprise the left-hand-side columns in Figure 4. The presence of image-based datasets `MNIST` and `CIFAR10` in this cluster is not surprising, as for superpixel graphs the structure loosely follows a grid layout for all classes, meaning determining class solely based on structure is difficult. Additionally, the coordinate information of superpixels is encoded also in the node features, together with average pixel intensities. A model with powerful enough classifier component is then sufficient for achieving high accuracy using these node features alone. Furthermore, the sensitivity of these datasets to *MidPass* and *HighPass* indicates that the overall shape of the signals encoded by low-frequencies is more informative for classifying the image content than sharp superpixel transitions encoded by high-frequencies. The presence of `ENZYMES` in `I-1` is likely due to the fact that some of the node features are precomputed using graph kernels, and therefore are sufficient to distinguish the enzyme classes in the dataset when structural information is removed. Last but not least, `PCQM4Mv2-subset` dataset appears to have a complex task that is dominated by the node feature information, yet the graph structure encodes non-negligible information as well. Out of all datasets in the `I-1` cluster, `PCQM4Mv2-subset` is the most sensitive one to structural perturbations. This corroborates with the expectation that predicting the HOMO-LUMO gap, which is the energy difference between the highest occupied molecular orbital (HOMO) and lowest unoccupied molecular orbital (LUMO), is a complex task that heavily depends on atom types, their bonds, and relative distances.

`I-2`**: Node features contain majority of necessary structural information.** For datasets in `I-2`, the graph structural information is again not necessary for achieving the baseline performance if the original node features are present, while the performance deteriorates noticably if *NoNodeFtrs* is applied. However, unlike `I-1`, these datasets are much less affected overall by the perturbations on node features. Many of the node features on these datasets are themselves derived from the graph's geometry, and it seems MPNNs are able to use either the graph structure or the node features to compensate for the absence of the other when encountering perturbed graphs. It appears that the low/mid/high-pass filterings in particular are able to retain a significant amount of geometric information.

The synthetic graphs of `Scale-Free` and `Small-world` (both `I-2` datasets) are generated through different algorithms (WS and BA, respectively), but the node features and tasks are equivalent: The features are the local clustering coefficient and PageRank score of each node and the task is to classify graphs based on average path length. Since the encoded features are derived from graph structure itself, MPNNs are still able to exploit them when the original graph structure is perturbed. When the MPNNs are forced to rely on graph structure instead, they are still able to attain AUROCs above random despite some decrease.

For many of the `I-2` datasets, *NodeDeg* allows one to replace geometric information of original node features with new geometric information, the degree of each vertex, to large success – for some of them the original AUROC scores are recovered and even surpassed, possibly due to *NodeDeg* reinforcing the existing structural signal. This trend is not as pronounced when the GIN-based model is used, since GIN achieves a comparatively high level of performance even in the face of *NoNodeFtrs*, likely due to the higher expressiveness of GIN compared to GCN in distinguishing of structural patterns.

On the other hand, there are datasets of biochemical origin in this cluster, whose node features encode chemical and physical attributes, such as atom or amino acid type. Except `MUTAG`, there appears to be some information encoded in these node features that is irreplaceable by graph structure or node degree information.

`I-3`**: Graph-structure reliance.** The `I-3` cluster is characterized by strong structural dependencies, and can be further divided into two subgroups based on their sensitivities to node feature perturbations.

The first subgroup, which consists of `PATTERN`, `COLLAB`, `IMDB-BINARY` and `REDDIT`, is not affected by node feature perturbations. These datasets do not have any original informative node features and their tasks appear to be purely structure-based. Indeed, in the case of `PATTERN` the task is to detect structural patterns in graphs, rendering node features irrelevant for the task. On the other

hand, structural perturbations such as *NoEdges* and *FullyConn* cause drastic performance drops in this group, since most of its task signals are sourced from graph structures. This group also exhibits limited to no sensitivity towards *Frag-k2* and *Frag-k3* perturbations, which test for degrees of reliance on longer range interactions by limiting information propagation to $\{2, 3\}$ hops. We still see prominent sensitivity to *Frag-k1*, though, implying reliance on information from immediate neighbors. We can attribute the insensitivity for $k > 1$ to inherent graph properties for some of these datasets: For dense networks like `PATTERN` or ego-nets such as `IMDB-BINARY` and `COLLAB`, just 1 or 2 hops recover the original graph – for these graphs, the notion of long-range information does not exist.

The second `I-3` subgroup, formed by `NCI` datasets and `Synthie`, are the datasets that are notably affected by *all* perturbations. For `Synthie`, this sensitivity stems from its construction. The four synthetic classes in `Synthie` are formed by combinations of two distributions of graph structures and two distributions of node features – elimination of either leads to a *partial* collapse in the distinguishability of two classes. The `NCI` classification tasks, similarly to related bioinformatics datasets in `I-2`, show a degree of reliance on the high-dimensional node features, but additionally, they are also dependent on non-local structure as they are among the datasets most adversely affected by *Frag-k2* and *Frag-k3*.

Synthetic datasets `CLUSTER` and `SYNTHETICnew` are also adversely affected by both structural and node feature perturbations. However, they stand out due to the magnitude of this effect. Many of the perturbations lead to a major decrease in AUROC and close-to-random performance. A closer inspection can provide an explanation. The task of `CLUSTER` is semi-supervised clustering of unlabeled nodes into six clusters, and the true cluster labels are given as node features in only a single node per cluster. *NoEdges* and *FullyConn* remove the cluster structure altogether, while *NoNodeFtrs* and *NodeDeg* remove the given cluster labels, rendering the task unsolvable in either case. In `SYNTHETICnew`, the two classes are derived from a "base" graph by a class-specific edge rewiring and node feature permutation, hence either graph structure or node features should differentiate the classes. Despite such expectation, we observe that the original node features alone are not sufficient, as structure perturbations have detrimental impact on the prediction performance. On the other hand GIN and GCN with *NodeDeg* can learn to distinguish the two classes even without the original node features. Thus, the original node features appear to be unnecessary, while after bandpass-filtering even provide misleading signal.

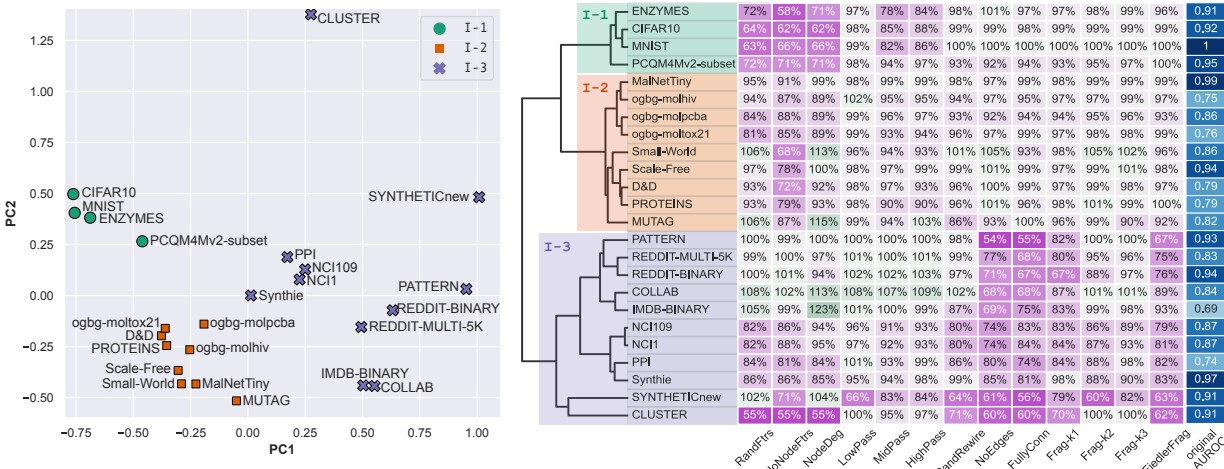

(a) Sensitivity profiles by **GCN** model (reprint of Figure 3a and 4).

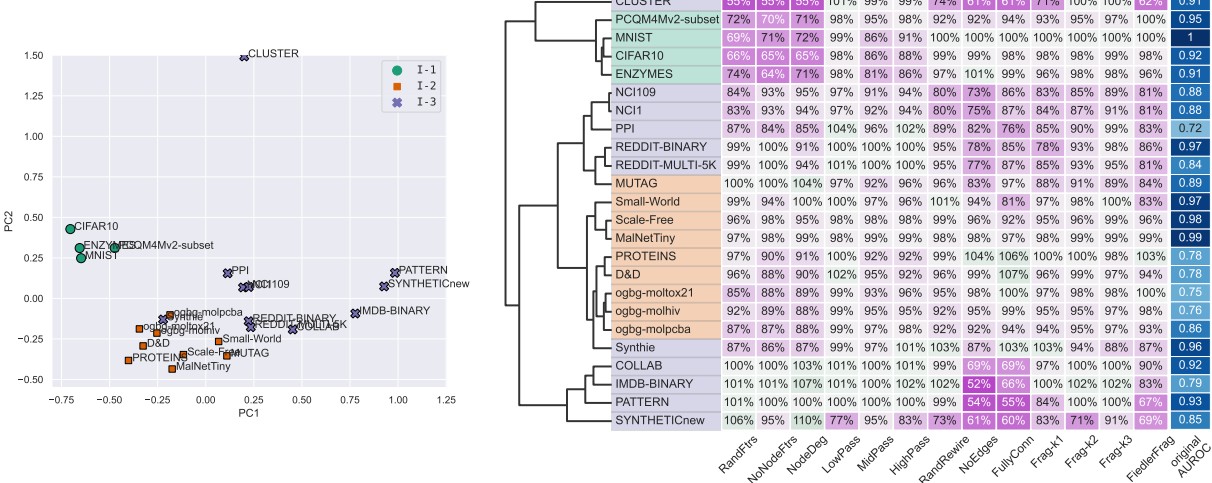

(b) Sensitivity profiles by **GIN** model; annotated by cluster assignment w.r.t. GCN model.

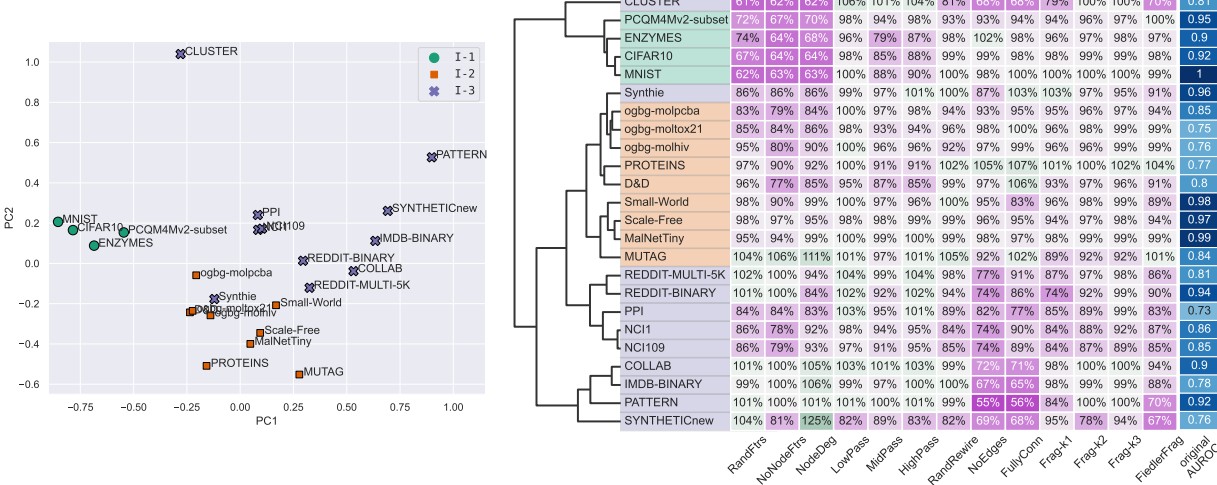

(c) Sensitivity profiles by **2-Layer GIN** model; annotated by cluster assignment w.r.t. GCN model.

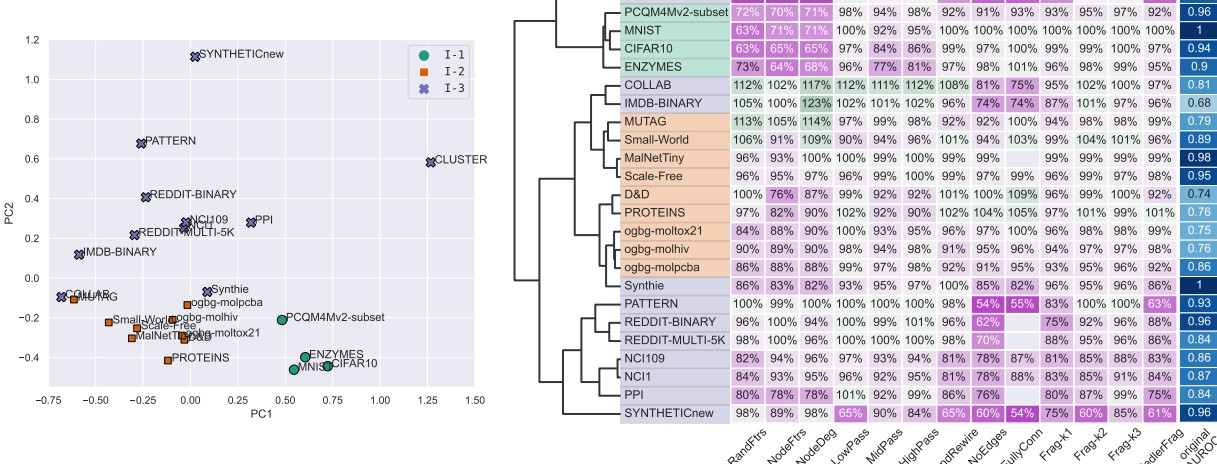

(d) Sensitivity profiles by **ChebNet** model; annotated by cluster assignment w.r.t. GCN model.

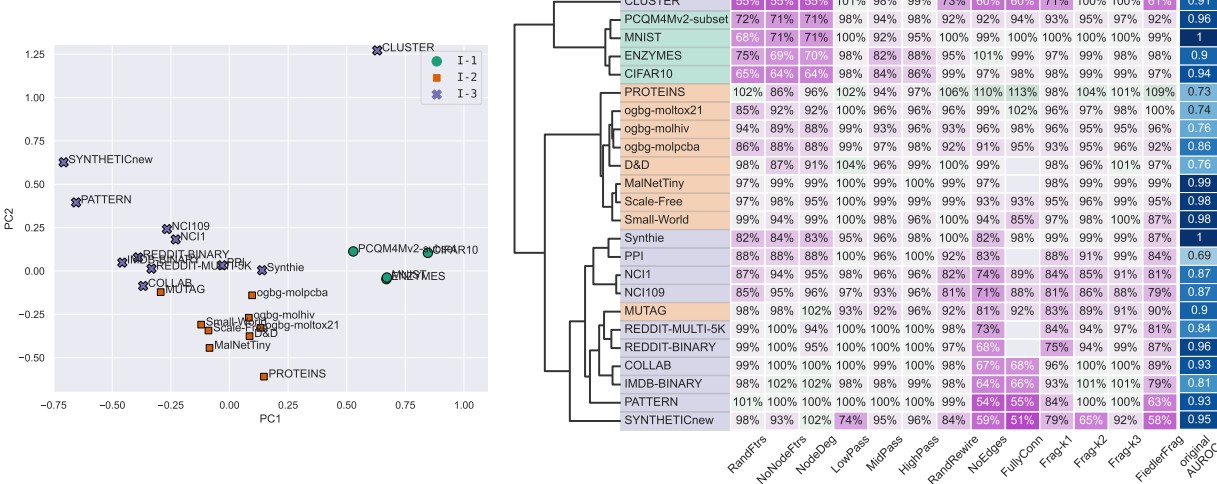

(e) Sensitivity profiles by **GatedGCN** model; annotated by cluster assignment w.r.t. GCN model.

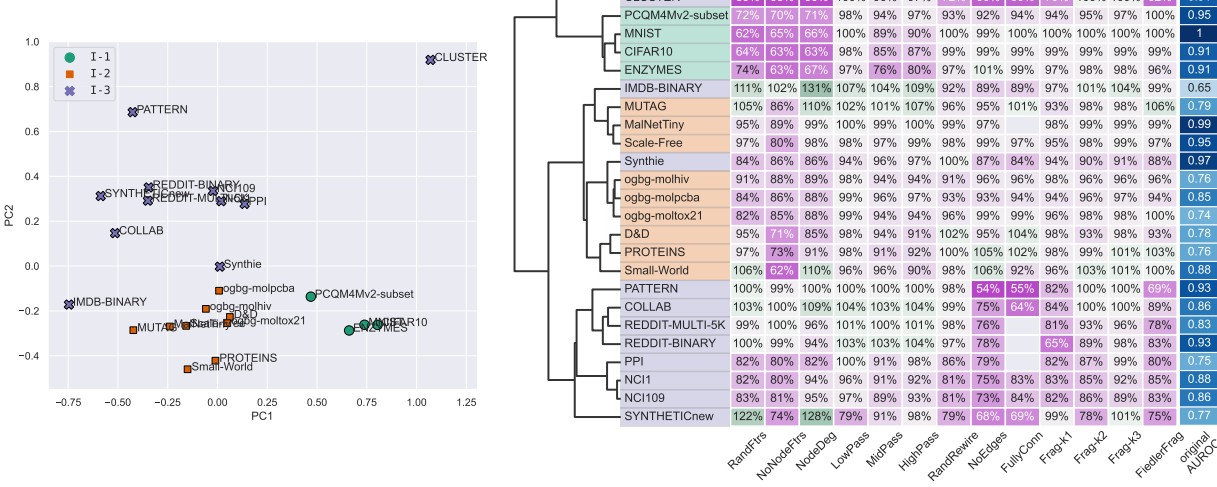

(f) Sensitivity profiles by **GCNII** model; annotated by cluster assignment w.r.t. GCN model.

**Figure B.1:** Taxonomy of inductive graph learning datasets via graph perturbations. The categorization into 3 dataset clusters is stable across the following models with only minor deviations: (a) GCN, (b) GIN, (c) 2-Layer GIN, (d) ChebNet, (e) GatedGCN, (f) GCNII. Panel (a) left and right is as shown in Figure 3a and 4, respectively, shown here for ease of comparison. Missing performance ratios (due to out-of-memory error) are shown in gray.

### B.2 Taxonomy of Transductive Benchmarks

**All transductive datasets are relatively insensitive to structural perturbations.** Unlike many of the inductive datasets that show significant reliance on the graph structure (`I-3`), the lowest performance achieved for a transductive dataset due to graph structure removal is still as high as 92% (`Flickr`), suggesting a weak dependence on the full graph structure. Furthermore, on average, considering only the neighborhoods of up to 3-hops (*Frag-k3*) nearly retains the full potential of the model (99% ± 1.6%), revealing the lack of long-range dependencies in these node-level datasets. Such negligence of the full graph structure might be attributed to the limitations of the GCN expressivity and issues such as oversquashing [59]. While these limitations are fundamentally true, our observation of long-range dependencies on some graph-level tasks like `NCI`, coupled with our architecture being 5 layers deep with residual connections, indicate that our GCN model is capable of capturing non-local information in the 3-hop neighborhoods. Furthermore, our observed long-range *independence* in transductive node-level datasets is consistent with the promising results presented by recent development of scalable GNNs that operate on subgraphs [12, 21, 70], breaking or limiting long-range connections.

`T-3`**: Indifference to node and structure removal.** The datasets in `T-3` are relatively insensitive to perturbations of graph structure and also to the removal of node features (*NoNodeFtrs* and *NodeDeg*). For example, the Amazon datasets (`Am-Phot` and `Am-Comp`) always achieve near perfect classification performance regardless of the perturbations applied, suggesting redundancy between node features and graph structure for the corresponding tasks. For these datasets, in particular, `GitHub`, `Am`, and `Twitch`, more sophisticated, or combinations of, perturbations might be needed to gauge their essential characteristics.

`T-2`**: Rich node features but substitutable for structural (summary) information.** `T-2` contains a broad spectrum of datasets from citation networks (`CF`), social networks (`Coau`, `FBPP`, `LFMA`), to web pages (`WikiNet`, `WikiCS`). The considerable performance decrease due to node feature removal suggests the relevance of the node features for their tasks. For example, it is not surprising that the binary bag-of-words features of `CF` datasets provide relevant information to classify papers into different fields of research, as one might expect some keywords to appear more likely in one field than in another. Furthermore, using the one-hot encoded node degrees (*NodeDeg*) always results in better performance over *NoNodeFtrs*. And in many cases such as Facebook (`FBPP`), *NodeDeg* nearly retains the baseline performance, suggesting the relevance of node degree information, as a form of structural summary, for the respective tasks.

`WebKB-Tex`, although clustered into `T-2` is more of an outlier that does not clearly fit into any of the existing clusters. As we will discuss more in `T-1`, `WebKB-Tex` considerably benefits from *HighPass*, while *LowPass* and *MidPass* severely decrease its performance.

`T-1`**: Heterophilic datasets.** Three of the four datasets in `T-1` (`Actor`, `WebKB-Cor`, and `WebKB-Wis`) are commonly referred to as heterophilic datasets [42, 48]. While `WebKB-Tex` (`T-2`) is also known to be heterophilic, it is isolated from `T-1` mainly due to its insensitivity to node feature removal, suggesting the structure alone is sufficient for its prediction task.

Our results show that in heterophilic datasets such as `T-1` and `WebKB-Tex`, *LowPass* node feature filtering, realized by local aggregation (Eq. 3), significantly degrades the performance, unlike other homophilic datasets. By contrast, *HighPass* results in better performance than *LowPass*. In the case of `WekbKB-Tex`, *HighPass* significantly improves the performance over the baseline. This observation is related to recent findings [42] that in the case of extreme heterophily, local information, this time in form of the neighborhood patterns, may suffice to infer the correct node labels.

Finally, despite heterophilic datasets [2, 42, 48, 59] attracting much recent attention, this type of datasets (`T-1` and `WebKB-Tex`) is lacking in availability compared to the others (`T-{2,3}`), which exhibit homophily but with different levels of reliance on node features. Thus, there is a need to collect and generate more real-world heterophilic datasets.

## B.3 Correlations of Perturbations

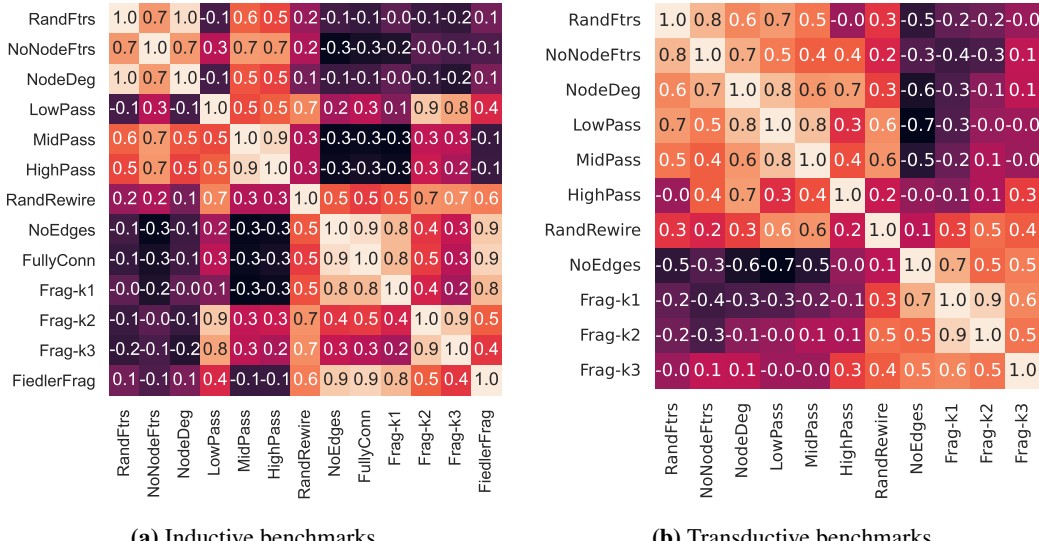

**(a)** Inductive benchmarks  **(b)** Transductive benchmarks

**Figure B.2:** Pearson correlation coefficients of the log2 performance fold change between different perturbations (w.r.t. a GCN model).

We compute the Pearson correlation between all pairs of perturbations based on the log2 performance fold change. The results in Figure B.2 indicate that many perturbations correlate with each other to some extend. For both transductive and inductive benchmarks, the perturbations roughly cluster into two groups, separating node feature perturbations (see Section 2.1) and graph structure perturbations (see Section 2.2). In particular, perturbations that replace the original node features with other less informative features, including *RandFtrs*, *NoNodeFtrs*, and *NodeDeg*, highly correlate with one another (Pearson $r \geq 0.6$). Similarly, perturbations that severely break the graphs apart, including *NoEdges*, *Frag-k1*, and *FiedlerFrag*, are highly correlated (Pearson $r \geq 0.8$).

# C  Graph Learning Benchmarks

## C.1  Inductive Datasets

MNIST **and** CIFAR10 [18] are derived from the well-known image classification datasets. The images are converted to graphs by SLIC superpixelization; node features are the average pixel coordinates and intensities; edges are constructed based on kNN criterion.

PATTERN **and** CLUSTER [18] are node-level inductive datasets generated from SBMs [31]. In PATTERN, the task is to identify nodes of a structurally specific subgraph; CLUSTER has a semi-supervised clustering task of predicting the true cluster assignment of nodes while observing only one labelled node per cluster.

IMDB-BINARY [68] is a dataset of ego-networks, where nodes represent actors/actresses and an edge between two nodes means that the two artists played in a movie together. The task is to determine which genre (action or romance) each ego-network belongs to.

D&D [17] is a protein dataset where each protein is represented by a graph with rich node feature set. The task is to classify proteins as enzymes or non-enzymes.

ENZYMES [6] is a dataset of tertiary structures from six enzymatic classes (determined by Enzyme Commission numbers). Each node represents a secondary structure element (SSE), and has an edge between its three spatially closest nodes. Node features are the type of SSE, and the physical and chemical information.

PROTEINS [6] is a modification of the D&D [17]; the task is the same but the protein graphs are generated as in ENZYMES.

NCI1 **and** NCI109 [61] consist of graph representations of chemical compounds; each graph represents a molecule in which nodes represent atoms and edges represent atomic bonds. Atom types are one-hot encoded as node features. The tasks are to determine whether a given compound is active or inactive in inhibiting non-small cell lung cancer (NCI1) or ovarian cancer (NCI109).

COLLAB [68] is an ego-network dataset of researchers in three different fields of physics. Each graph is a researcher's ego-network, where nodes are researchers and an edge between two nodes means the two researchers have collaborated on a paper. The task is to determine which field a given researcher ego-network belongs to.

REDDIT-BINARY **and** REDDIT-MULTI-5K [68] graphs are derived from Reddit communities (subreddits). These subreddits are Q&A based or discussion-based. Each graph represents a set of interactions between users through posts and comments; nodes represent users while an edge implies an interaction between two users. The task for REDDIT-BINARY is to determine whether the given interaction graph belongs to a Q&A or discussion subreddit. In REDDIT-MULTI-5K, the graphs are drawn from 5 specific subreddits instead, and the task is to predict the subreddit a graph belongs to.

MUTAG [15] is a dataset of Nitroaromatic compounds. Each compound is represented by a graph in which nodes represent atoms with their types one-hot encoded as node features, and edges represent atomic bonds. The task is to determine whether a given compound has mutagenic effects on Salmonella typhimurium bacteria.

MalNet-Tiny [23] is a smaller version of MalNet dataset, consisting of function call graphs of various malware on Android systems using Local Degree Profiles as node features. In MalNet-Tiny, the task is constrained to classification into 5 different types of malware.

ogbg-molhiv**,** ogbg-molpcba**,** ogbg-moltox21 [33] datasets, adopted from MoleculeNet [64], are composed of molecular graphs, where nodes represent atoms and edges represent atomic bonds in-between. Node features include atom type and physical/chemical information such chirality and charge. The task is to classify molecules on whether they inhibit HIV replication (ogbg-molhiv) or their toxicity on on 12 different targets such as receptors and stress response pathways in a multilabel classification setting (ogbg-moltox21). In ogbg-molpcba the task is 128-way multi-task binary classification derived from 128 bioassays from PubChem BioAssay.

PCQM4Mv2-subset is our derivative of the OGB-LSC PCQM4Mv2 [34] molecular dataset. The original task is a regression of a quantum physical property – the HOMO-LUMO gap. For compatibility with our analysis, we quantized the regression task into 20-way classification task based on quantils of the training set. As true labels of the original "test-dev" and "test-challange" dataset splits are kept private by the OGB-LSC challenge organizers, and for efficiency of our analysis, we created a custom reduced splits as follows: *train set*: random 10% of the original train set; *validation set*: another random 50,000 graphs from the original train set; *test set*: the original validation set. The molecular graphs are featurized the same way as in ogbg-mol* datasets.

PPI [30, 72] dataset contains a collection of 24 tissue-specific protein-protein interaction networks derived from the STRING database [58] using tissue-specific gold-standards from [26]. 20 of the networks are used for training, 2 used for validation, and 2 used for testing. In each network, each protein (node) is associated with 50 different gene signatures as node features. The multi-label node classification task was to classify each gene (node) in a graph based on its gene ontology terms.

SYNTHETICnew [19] is a dataset where each graph is based on a random graph $G$ with scalar node features drawn from the normal distribution. Two classes of graphs are generated from $G$ by randomly rewiring edges and permuting node attributes; the number of rewirings and permuted attributes are distinct for the two classes. Noise is added to the node features to make the tasks more difficult. The task is to determine which class a given graph belongs to.

Synthie [45] dataset is generated from two Erdös-Rényi graphs $G_{1,2}$: Two sets of graphs $S_{1,2}$ are then generated by randomly adding and removing edges from $G_{1,2}$. Then, 10 graphs were sampled from these sets and connected by randomly adding edges, resulting in a single graph. Two classes of these graphs, $C_{1,2}$ are generated by using distinct sampling probabilities for the two sets. The two classes are then in turn split into two by generating two sets of vectors $A$ and $B$; nodes from a given graph were appended a vector from $A$ as node features if they were sampled from $S_1$, and $B$ for $S_2$ for one class, and vice versa for the other. The task is to classify which of these four classes a given graph belongs to.

**Table C.1:** Inductive benchmarks. All datasets are equipped with graph-level classification tasks, except PATTERN and CLUSTER that are equipped with inductive node-level classification tasks.

| Dataset | # Graphs | Avg # Nodes | Avg # Edges | # Features | # Classes | Predef. split | Ref. |
|---|---|---|---|---|---|---|---|
| MNIST | 70,000 | 70.57 | 564.53 | 3 | 10 | Yes | [18] |
| CIFAR10 | 60,000 | 117.63 | 941.07 | 5 | 10 | Yes | [18] |
| PATTERN | 14,000 | 118.89 | 6,078.57 | 3 | 2 | Yes | [18] |
| CLUSTER | 12,000 | 117.20 | 4,301.72 | 7 | 6 | Yes | [18] |
| IMDB-BINARY | 1,000 | 19.77 | 96.53 | – | 2 | No | [68] |
| D&D | 1,178 | 284.32 | 715.66 | 89 | 2 | No | [17] |
| ENZYMES | 600 | 32.63 | 62.14 | 21 | 6 | No | [6] |
| PROTEINS | 1,113 | 39.06 | 72.82 | 4 | 2 | No | [6] |
| NCI1 | 4,110 | 29.87 | 32.3 | 37 | 2 | No | [61] |
| NCI109 | 4,127 | 29.68 | 32.13 | 38 | 2 | No | [61] |
| COLLAB | 5,000 | 74.49 | 2,457.78 | – | 3 | No | [68] |
| REDDIT-BINARY | 2,000 | 429.63 | 497.75 | – | 2 | No | [68] |
| REDDIT-MULTI-5K | 4,999 | 508.52 | 594.87 | – | 5 | No | [68] |
| MUTAG | 188 | 17.93 | 19.79 | 7 | 2 | No | [15] |
| MalNet-Tiny | 5,000 | 1,410.3 | 2,859.94 | 5 | 5 | No | [23] |
| ogbg-molhiv | 41,127 | 25.5 | 27.5 | 9 sets | 2 | Yes | [33] |
| ogbg-molpcba | 437,929 | 26.0 | 28.1 | 9 sets | 128x binary | Yes | [33] |
| ogbg-moltox21 | 7,831 | 18.6 | 19.3 | 9 sets | 12x binary | Yes | [33] |
| PCQM4Mv2-subset | 446,405 | 14.1 | 14.6 | 9 sets | quantized to 20 | Custom | [34] |
| PPI | 24 | 2,372.67 | 66,136 | 50 | 121 | Yes | [72] |
| SYNTHETICnew | 300 | 100 | 196 | 1 | 2 | No | [19] |
| Synthie | 400 | 95 | 196.25 | 15 | 4 | No | [45] |
| Small-world | 256 | 64 | 694 | 2 | 10 | No | [69] |
| Scale-free | 256 | 64 | 501.56 | 2 | 10 | No | [69] |

Small-world **and** Scale-free [69] datasets are generated by tweaking graph generation parameters for the real-world-derived small-world [63] and scale-free [32] graphs. Graphs are generated using a range of Averaging Clustering Coefficient and Average Path Length parameters. In our experiments, clustering coefficients and PageRank scores constitute node features while the task is to classify graphs based on average path length, where the continuous path length variable is rendered discrete by 10-way binning.

## C.2 Transductive Node-level Datasets

WikiNet [51] contains two networks of Wikipedia pages, where edges indicate mutual links between pages, and node features are bag-of-words (BOW) of informative nouns. The task is to classify the web pages based on their average monthly traffic bins.

WebKB [51] contains networks of web pages from different universities, where an (directed) edge is a hyperlink between two web pages, with BOW node features. The task is to classify the web pages into five categories: student, project, course, staff, and faculty.

Actor [51] is a network of actors, where an edge indicate co-occurrence of two actors on a same Wikipedia page, with node features represented by keywords about the actor on Wikipedia. The task is to classify the actor into one of five categories.

WikiCS [43] is a network of Wikipedia articles related to Computer Science, where edges represent hyperlinks between them, with 300-dimensional word embeddings of the articles. The task is to classify the articles into one of ten branches of the field.

Flickr [70] is a network of images, where the edges represent common properties between images, such as locations, gallery, and comments by the same users. The node features are BOW of image descriptions, and the task is to predict one of 7 tags for an image.

CF (CitationFull) [5] contains citation networks where nodes are papers and edges represent citations, with node features as BOW of papers. The task is to classify the papers based on their topics.

DzEu (DeezerEurope) [53] is a network of Deezer users from European countries where nodes are the users and edges are mutual follower relationships. The task is to predict the gender of users.

**Table C.2:** Transductive benchmarks with node-level classification tasks.

| Dataset | # Nodes | # Edges | # Node feat. | # Pred. classes | Predef. split | Ref. |
|---|---|---|---|---|---|---|
| WikiNet-cham | 2,277 | 72,202 | 128 | 5 | Yes | [51] |
| WikiNet-squir | 5,201 | 434,146 | 128 | 5 | Yes | [51] |
| WebKB-Cor | 183 | 298 | 1,703 | 10 | Yes | [51] |
| WebKB-Wis | 251 | 515 | 1,703 | 10 | Yes | [51] |
| WebKB-Tex | 183 | 325 | 1,703 | 10 | Yes | [51] |
| Actor | 7,600 | 30,019 | 932 | 10 | Yes | [51] |
| WikiCS | 11,701 | 297,110 | 300 | 10 | Yes | [43] |
| Flickr | 89,250 | 899,756 | 500 | 7 | Yes | [70] |
| CF-Cora | 19,793 | 126,842 | 8,710 | 70 | No | [5] |
| CF-CoraML | 2,995 | 16,316 | 2,879 | 7 | No | [5] |
| CF-CiteSeer | 4,230 | 10,674 | 602 | 6 | No | [5] |
| CF-DBLP | 17,716 | 105,734 | 1,639 | 4 | No | [5] |
| CF-PubMed | 19,717 | 88,648 | 500 | 3 | No | [5] |
| DzEu | 28,281 | 185,504 | 128 | 2 | No | [53] |
| LFMA | 7,624 | 55,612 | 128 | 18 | No | [53] |
| Am-Comp | 13,752 | 491,722 | 767 | 10 | No | [56] |
| Am-Phot | 7,650 | 238,162 | 745 | 8 | No | [56] |
| Coau-CS | 18,333 | 163,788 | 6,805 | 15 | No | [56] |
| Coau-Phy | 34,493 | 495,924 | 8,415 | 5 | No | [56] |
| Twitch-EN | 7,126 | 77,774 | 128 | 2 | No | [52] |
| Twitch-ES | 4,648 | 123,412 | 128 | 2 | No | [52] |
| Twitch-DE | 9,498 | 315,774 | 128 | 2 | No | [52] |
| Twitch-PT | 1,912 | 64,510 | 128 | 2 | No | [52] |
| Github | 37,700 | 578,006 | 128 | 2 | No | [52] |
| FBPP | 22,470 | 342,004 | 128 | 4 | No | [52] |

LFMA (LastFMAsia) [53] is a network of LastFM users from Asian countries where edges are mutual follower relationships between them. The task is to predict the location of users.

Amazon [56] contains Amazon Computers and Amazon Photo. They are segments of the Amazon co-purchase graph, where nodes represent goods, edges indicate that two goods are frequently bought together, node features are bag-of-words encoded product reviews, and class labels are given by the product category.

Coau (Coauthor) [56] contains Coauthor CS and Coauthor Physics. They are co-authorship graphs based on the Microsoft Academic Graph from the KDD Cup 2016 challenge 3. Nodes are authors, and are connected by an edge if they co-authored a paper; node features represent paper keywords for each author's papers, and class labels indicate most active fields of study for each author.

Twitch [52] contains Twitch user-user networks of gamers who stream in a certain language where nodes are the users themselves and the edges are mutual friendships between them. The task is to to predict whether a streamer uses explicit language. Due to low baseline performance even after a thorough hyperparameter search, we excluded Twitch-RU and Twitch-FR from our main analysis.

Github [52] is a network of GitHub developers where nodes are developers who have starred at least 10 repositories and edges are mutual follower relationships between them. The task is to predict whether the user is a web or a machine learning developer.

FBPP (FacebookPagePage) [52] is a network of verified Facebook pages that liked each other, where nodes correspond to official Facebook pages, edges to mutual likes between sites. The task is multi-class classification of the site category.

# D  Distribution of Classical Graph Properties in Benchmarking Datasets

In this work we use *perturbation sensitivity profiles* derived from a GNN's prediction performance in order to gauge *how task-related information* is encoded in the graph datasets. In this section we explore an alternative approach. We analyze classical graph properties in multiple datasets and their classes to investigate whether we can establish a meaningful taxonomy without any dependence on a particular GNN method, while using well-established graph properties.

**Table D.1:** Classical graph properties among positive and negative classes of 9 graph-classification datasets. The difference between datasets dominates within-dataset differences between classes.

| | Num. nodes | Num. edges | Density | Connectivity | Diameter | Approx. max clique | Centrality | Cluster. coeff. | Num. triangles |
|---|---|---|---|---|---|---|---|---|---|
| IMDB-BINARY (class=0) | 20.11 | 96.78 | 0.559 | 3.828 | 1.838 | 10.30 | 0.559 | 0.943 | 307.73 |
| IMDB-BINARY (class=1) | 19.43 | 96.29 | 0.482 | 3.388 | 1.884 | 10.01 | 0.482 | 0.951 | 476.25 |
| REDDIT-BINARY (class=0) | 641.25 | 735.95 | 0.012 | 0.556 | 5.646 | 3.22 | 0.012 | 0.054 | 35.96 |
| REDDIT-BINARY (class=1) | 218.00 | 259.56 | 0.032 | 0.423 | 3.778 | 2.95 | 0.032 | 0.041 | 13.71 |
| D&D (class=0) | 341.88 | 870.23 | 0.019 | 1.110 | 20.843 | 4.95 | 0.019 | 0.479 | 617.07 |
| D&D (class=1) | 183.72 | 449.43 | 0.040 | 1.140 | 17.460 | 4.79 | 0.040 | 0.480 | 302.55 |
| PROTEINS (class=0) | 50.00 | 94.06 | 0.142 | 1.196 | 13.837 | 3.85 | 0.142 | 0.473 | 34.30 |
| PROTEINS (class=1) | 22.94 | 41.52 | 0.315 | 1.420 | 7.278 | 3.80 | 0.315 | 0.575 | 17.24 |
| NCI1 (class=0) | 25.65 | 27.65 | 0.100 | 0.924 | 11.265 | 2.02 | 0.100 | 0.002 | 0.03 |
| NCI1 (class=1) | 34.07 | 36.94 | 0.078 | 0.796 | 11.917 | 2.05 | 0.078 | 0.004 | 0.07 |
| NCI109 (class=0) | 25.61 | 27.61 | 0.100 | 0.913 | 11.061 | 2.02 | 0.100 | 0.002 | 0.02 |
| NCI109 (class=1) | 33.69 | 36.59 | 0.079 | 0.794 | 11.644 | 2.05 | 0.079 | 0.004 | 0.07 |
| MUTAG (class=0) | 13.94 | 14.62 | 0.169 | 1.000 | 7.016 | 2.00 | 0.169 | 0.000 | 0.00 |
| MUTAG (class=1) | 19.94 | 22.40 | 0.123 | 1.000 | 8.824 | 2.00 | 0.123 | 0.000 | 0.00 |
| SYNTHETICnew (class=0) | 100.00 | 196.42 | 0.040 | 0.993 | 7.333 | 3.00 | 0.040 | 0.024 | 5.39 |
| SYNTHETICnew (class=1) | 100.00 | 196.08 | 0.040 | 0.993 | 7.213 | 3.00 | 0.040 | 0.022 | 4.54 |
| ogbg-molhiv (class=0) | 25.20 | 27.13 | 0.104 | 0.931 | 11.016 | 2.02 | 0.104 | 0.002 | 0.03 |
| ogbg-molhiv (class=1) | 34.18 | 36.69 | 0.084 | 0.824 | 12.183 | 2.01 | 0.084 | 0.001 | 0.01 |

A static analysis of the graph properties alone is insufficient without taking into account the prediction task as well. The graph domain that a dataset $X$ is sampled from (e.g., drug-like molecules, proteins, ego networks, citation networks) may exhibit varying range of properties (e.g., density, node degree distribution, local/global clustering coefficients, number of triangles, graph diameter, girth, maximum clique, etc.), however these do not take into account node features in attributed graphs, and could be irrelevant to the prediction task $Y$. Therefore, we look at the difference in graph properties compared among the individual classes of $Y$.

Particularly, we look at all 9 inductive binary-classification datasets from our dataset selection (Table C.1). Within each class (the negative and positive label) of these 9 datasets we computed the average value of 9 graph properties computed by the NetworkX package [27]. The results are presented in Table D.1 and Figure D.1. Primarily, the computed graph properties vary more between datasets than between classes. The marginal graph properties of the positive and negative class are very similar to each other, especially for the SYNTHETICnew dataset. The largest difference between the classes appears to be the average size of the graphs, which is captured by the average number of nodes and edges. Therefore we argue that basing a taxonomy on dataset or class-level marginal graph properties is grossly insufficient as it completely fails to capture the nature of the prediction task.

Alternatively, one could conduct a correlation analysis between classical graph properties (averaged per class) and the outcome $Y$. However, that would again only take into account the marginal properties, assume linear relationship (as correlation captures only a linear relationship), and would rely on a fixed set of computable graph properties. These appear to be fundamental limitations compared to the perturbation analysis presented in the main text, that would result in a grossly skewed taxonomy.

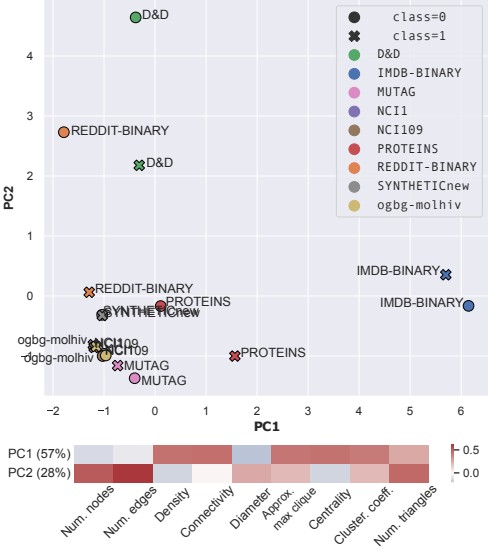

**Figure D.1:** PCA plot of 9 binary graph-level classification datasets represented by their per-class graph properties. In the bottom, the loadings of the first two principal components are shown.

# E   Impact of random initialization on *Frag-k* perturbations

Our *Frag-k* perturbation is potentially sensitive to the random initializations of the initial seed nodes used in the fragmentation procedure. To measure this sensitivity of *Frag-k* perturbations to node initializations, we computed the variance of AUROC results across ten experiments with different random seeds for both GCN and GIN models. Here, we analysed five datasets, the performance on which was significantly altered by *Frag-k* in the original analysis, namely, CLUSTER, PATTERN, PPI, Synthie, and SYNTHETICnew. The variances are within 5%, with the only exception being SYNTHETICnew. We hypothesize that this is due to the randomness of the constructions of the SYNTHETICnew dataset. Thus, overall, the *Frag-k* approach is sufficiently stable for datasets whose constructions involve little randomness.

**Table E.1:** Variances of AUROC across ten different random seeds for *Frag-k* for GCN.

| Dataset | Perturbation | AUROC Avg. | AUROC Std. | AUC Std./Avg. (%) |
|---|---|---|---|---|
| CLUSTER | *Frag-k1* | 0.637 | 0.001 | 0.165 |
| CLUSTER | *Frag-k2* | 0.913 | 0.000 | 0.039 |
| CLUSTER | *Frag-k3* | 0.913 | 0.000 | 0.037 |
| PATTERN | *Frag-k1* | 0.769 | 0.001 | 0.095 |
| PATTERN | *Frag-k2* | 0.933 | 0.000 | 0.016 |
| PATTERN | *Frag-k3* | 0.933 | 0.000 | 0.021 |
| PPI | *Frag-k1* | 0.620 | 0.003 | 0.529 |
| PPI | *Frag-k2* | 0.647 | 0.012 | 1.807 |
| PPI | *Frag-k3* | 0.720 | 0.011 | 1.519 |
| SYNTHETICnew | *Frag-k1* | 0.704 | 0.126 | 17.908 |
| SYNTHETICnew | *Frag-k2* | 0.533 | 0.078 | 14.701 |
| SYNTHETICnew | *Frag-k3* | 0.715 | 0.089 | 12.492 |
| Synthie | *Frag-k1* | 0.962 | 0.015 | 1.581 |
| Synthie | *Frag-k2* | 0.870 | 0.029 | 3.334 |
| Synthie | *Frag-k3* | 0.876 | 0.036 | 4.164 |

**Table E.2:** Variances of AUROC across ten different random seeds for *Frag-k* for GIN.

| Dataset | Perturbation | AUROC Avg. | AUROC Std. | AUC Std./Avg. (%) |
|---|---|---|---|---|
| CLUSTER | *Frag-k1* | 0.643 | 0.001 | 0.162 |
| CLUSTER | *Frag-k2* | 0.910 | 0.001 | 0.101 |
| CLUSTER | *Frag-k3* | 0.910 | 0.001 | 0.130 |
| PATTERN | *Frag-k1* | 0.780 | 0.001 | 0.091 |
| PATTERN | *Frag-k2* | 0.934 | 0.000 | 0.013 |
| PATTERN | *Frag-k3* | 0.934 | 0.000 | 0.019 |
| PPI | *Frag-k1* | 0.617 | 0.002 | 0.376 |
| PPI | *Frag-k2* | 0.644 | 0.009 | 1.476 |
| PPI | *Frag-k3* | 0.704 | 0.013 | 1.843 |
| SYNTHETICnew | *Frag-k1* | 0.708 | 0.081 | 11.407 |
| SYNTHETICnew | *Frag-k2* | 0.532 | 0.071 | 13.276 |
| SYNTHETICnew | *Frag-k3* | 0.757 | 0.064 | 8.411 |
| Synthie | *Frag-k1* | 0.985 | 0.008 | 0.810 |
| Synthie | *Frag-k2* | 0.945 | 0.011 | 1.213 |
| Synthie | *Frag-k3* | 0.920 | 0.025 | 2.677 |

