# OpenReview forum: "Taxonomy of Benchmarks in Graph Representation Learning"
_logconference.io/LOG/2022/Conference — LoG 2022 Oral_

### Official Review · Reviewer_xDho · 2022-10-13

**Overall Score:** 6
**Confidence:** 4

**Review:**

The authors systematically study the taxonomy of benchmark datasets in graph representation learning. They utilize sensitivity profiles to probe to what extent how much GNN performance changes due to node feature and graph structure perturbations. They conduct extensive experiments to study the categorization of graph learning datasets on the commonly-used small- and middle-scale datasets under inductive and transductive benchmarks.

 I reviewed this paper before, and the authors improved the manuscript and the paper reads much more nicely now. Overall speaking, the empirical results of the paper are good. As points for future improvement, I believe that further investigations of more categorizations of graph datasets and the analysis on large-scale datasets would lead to a much stronger paper.

**Pros:**

- The idea of taxonomizing datasets in graph representation learning is interesting.
- The authors conduct extensive experiments to show the sensitivity profile of each graph dataset.

**Cons:**

- This paper limits the taxonomy of graph datasets in the sensitivity profile. I suggest the authors could incorporate more graph characteristics, such as degree and cluster coefficient, to improve the versatility and usefulness of the proposed method.
- The authors report results in the paper on the small- and middle-scale datasets. I think the author should conduct experiments on large-scale graph datasets, such as the open graph benchmark (OGB-LSC) benchmark [1]. Because when the size of the graph dataset increases, the intrinsic property of the graph dataset changes greatly.

[1] Chen M, Wei Z, Huang Z, et al. Simple and deep graph convolutional networks[C]//International Conference on Machine Learning. PMLR, 2020: 1725-1735.

---

### Official Review · Reviewer_2Pth · 2022-10-17

**Overall Score:** 6
**Confidence:** 4

**Review:**

### Summary
In this paper, the authors propose GTaxoGym, a taxonomy approach of benchmarks in graph representation learning. In particular, this paper aims to generate dataset sensitivity profiles concerning the importance of node features and graph structures (or node connections). To achieve this, they propose to use six different node perturbations and seven different graph structure perturbations to form corresponding dataset sensitivity profiles, with a GCN model.

### Pros:
1. The way they generate dataset sensitivity profiles is interesting and informative.
2. The robustness of these proposed perturbation methods is justified by using different GNN architectures.

### Cons:
1. The writing of this paper is not that good and is a little hard to follow. I recommend authors further polish the paper.
2. There are some typos. For example, they use commas at the end of the sentences in section one when they list the contributions.
3. The clustering results of different GNN models are missing. Do they have the same or different clustering results?

---

### Official Review · Reviewer_Gibg · 2022-10-21

**Overall Score:** 8
**Confidence:** 4

**Review:**

*Contributions*

The paper presents a study on datasets used to evaluate graph (node) classification. Benchmarks for graph classification methods have been done, and in past few years a range of new graph (collections) have been introduced for the purpose of benchmarking. However, what the important properties are in those graphs on which the classification is based is not well studied and only understood to a limited extent. This is exactly where the paper is situated.

The approach taken is that perturbation methods are designed to make large changes to the graphs, and then the performance of a graph classifier is compared between the original graph(s) and the perturbed graph(s). Specifically, their AUCs are compared and the used score is the log2 of the relative scores, such that (in expectation; assuming no inverse correlations will appear) a normalized score between -1 and +1 is obtained. The vector of these scores for a dataset is called its 'sensitivity profile'. These scores a listed in a large table, both PCA-based scatter plots of these results and a hierarchical clustering over the datasets is presented and analysed.

There are several findings: at the top level the datasets cluster into three groups, which do not align with the domains where the graphs come from. It is shown the synthetic datasets are far away from the real data, so in an evaluation of new methods, real data should be included. The authors suggest a representative set of graphs that could be used in future evaluations. Finally, it is pointed out the evaluation may depend on the choice of classifier method and beyond GCN and GIN that were the main methods, it is considered whether the dataset statistics would vary if other GNN methods are used, but it appears to be quite robust.

*Strong points*
- The work and approach are novel and they target a very important problem
- The paper is well-presented and easy to follow, with clear motivation
- The analysis is thorough and there are clear and possibly actionable results
- The introduced methodology could also be applied to other settings

*Weak points*
- There are possibly many open ends, it is difficult to assess whether the results as a whole a robust: the impact of various perturbations is not necessarily on the same scale. Hence, one could wonder whether the AUC scores after the perturbations are really comparable. If not, it does not make sense to apply a clustering method on top. Besides, it is not evaluated whether the use of very different type of methods (not GNNs) would yield the similar results.
- The paper violates the submission guidelines as it refers to the code repository, which is not anonymous. It could (should?) be desk rejected for this reason.

*Recommendation*

The paper is solid, well-presented, and contains new insights into a very important topic. They are limited in scope (see weak points), but they are a great starting point. I recommend for the paper to be accepted.

*Minor comments*

I am not sure the constructed tree should be called a 'taxonomy', because to me that would require explaining each split in an interpretable manner and making sure the hierarchy is solid. That is currently not done, only the top-level relations are analyzed and it appears to me that using different methods would impact the details of the hierarchy potentially quite a lot. It would be better to phrase this more appropriately.

---

### Official Review · Reviewer_DtB4 · 2022-10-21

**Overall Score:** 6
**Confidence:** 5

**Review:**

This paper proposes a data-driven approach to investigate what types of modeling abilities are required for a GNN model to perform well on each benchmark dataset. In particular, the proposed approach generates a sensitivity profile for each benchmark dataset that consists of the GNN performance after perturbing the dataset in different ways. Clustering can also be applied to the sensitivity profiles for different datasets to generate a taxonomy of the datasets.

---

Strength:

1) The data-driven approach to taxonomize benchmark datasets is a creative and interesting idea, and it could be potentially helpful for the future development of GNN methods.

2) Constructing a sensitivity profile by perturbing the dataset generally makes much sense as a way to characterize the datasets.

3) This paper conducts a relatively large scale experiments to implement and verify the proposed approach.

4) The paper is very well-written and easy-to-follow. I especially appreciate Figure 1, which is an excellent illustration of the proposed method.

---

Weakness and suggestions:

1) One of my major suggestions is about the choices and intuitions about the perturbations. In general, more intuitions should be provided for what exactly is each perturbation capturing.

    1.1) The study includes two ways to completely remove the node features, NoNodeFtrs and RandFtrs. Could the authors elaborate on what is the difference between these two perturbations? For example, if the decrease of accuracy differs significantly on these two perturbations, what does it mean?

    1.2) The presentation of the spectral node feature perturbations could be improved. To better align with the purpose of this paper, the presentation should focus on the high-level intuitions about what information do the low-, mid-, and high-band pass filtered node features capture (which is largely missing in the current submission), instead of the mathematical details of the construction of these filters. Since these filters are well-known techniques in the literature of graph signal processing, instead of the contribution of this work, the mathematical details could be put into Appendix, with a citation to a tutorial of graph signal processing.

    1.3) The design of the "Frag-k" perturbation is a bit ad-hoc. Mature community detection methods or graph partitioning methods could better serve the purpose of distinguishing the local vs global structure.

    1.4) Again for the "Frag-k" perturbation, how sensitive is the result with respect to the random choices of the "seed nodes"?

2) The sensitivity profile is constructed by the "log2-transformed ratio of test AUROC". The choice of "log2-transformed ratio" is a very critical design choice for the proposed method and should be better justified.

    2.1) Why does this metric make sense?

    2.2) How sensitive are the conclusions/findings with respect to different choices of the metric? For example, log10 or natural log instead of log2? Ratio or difference of AUROC without log transformed? Additional sensitivity analysis would be appreciated. A sensitivity analysis like this does not require retraining the GNNs and should be feasible if the AUROC scores were stored from the previous experiments.

3) The study includes 6 synthetic datasets where "... we have concrete understanding of their graph domain properties and how these properties relate to their prediction task". This is a great way to verify whether the proposed approach can reasonably cluster the benchmark datasets. However, **there is little analysis on the results of synthetic datasets**. If I didn't miss anything, the only result analysis about the synthetic datasets is "synthetic datasets do not fully represent real-world scenarios". And looking at the clustering results of the synthetic datasets, I do not see a clear pattern that aligns well with the nature of these synthetic datasets. This casts a doubt on whether the proposed sensitivity profile really captures (and to which extent) the essence of different benchmark datasets.

---

Overall, this paper is well-written with a very intriguing idea of data-driven approach for building a taxonomy of benchmark datasets. However, some design choices of the implementation seem to be ad-hoc and the empirical results are not convincing enough that the current implementation really captures the essence of the benchmark datasets.

---

### Meta-Review · Area_Chair_peov · 2022-11-18

**Confidence:** 4
**Recommendation:** Accept for spotlight

**Meta Review:**

This paper studies what aspects of a graph dataset influences the ability of GNNs to extract useful information from it. It takes a data-driven approach based on a sensitivity profile, that measures changes in performance under several perturbations on the data.
Reviewers were unanimous to find this submission highly interesting and tackling an important problem. Authors engaged with reviewers during the rebuttal process, resulting in an even stronger submission. The AC agrees with the assessment and therefore recommends acceptance at this point.

---

### Decision · Program_Chairs · 2022-11-23

**Decision:**

Accept (Oral)

**Comment:**

Congrats. There seems to be full agreement between the AC and reviewers to accept the work!